# Two SEPALLATA MADS-Box Genes, *SlMBP21* and *SlMADS1*, Have Cooperative Functions Required for Sepal Development in Tomato

**DOI:** 10.3390/ijms25052489

**Published:** 2024-02-20

**Authors:** Jianling Zhang, Zongli Hu, Qiaoli Xie, Tingting Dong, Jing Li, Guoping Chen

**Affiliations:** 1Laboratory of Plant Germplasm Innovation and Utilization, School of Life Sciences, Liaocheng University, Liaocheng 252000, China; zhangjianling0520@126.com; 2Laboratory of Molecular Biology of Tomato, Bioengineering College, Chongqing University, Chongqing 400030, China; qiaolixie@cqu.edu.cn (Q.X.); dtt@jsnu.edu.cn (T.D.); micy180605@163.com (J.L.); 3Institute of Integrative Plant Biology, School of Life Science, Jiangsu Normal University, Xuzhou 221116, China

**Keywords:** MADS-box, tomato, *SlMBP21*, *SlMADS1*, sepal size, fusion, cooperative functions

## Abstract

MADS-box transcription factors have crucial functions in numerous physiological and biochemical processes during plant growth and development. Previous studies have reported that two MADS-box genes, *SlMBP21* and *SlMADS1,* play important regulatory roles in the sepal development of tomato, respectively. However, the functional relationships between these two genes are still unknown. In order to investigate this, we simultaneously studied these two genes in tomato. Phylogenetic analysis showed that they were classified into the same branch of the SEPALLATA (SEP) clade. qRT-PCR displayed that both *SlMBP21* and *SlMADS1* transcripts are preferentially accumulated in sepals, and are increased with flower development. During sepal development, *SlMBP21* is increased but *SlMADS1* is decreased. Using the RNAi, tomato plants with reduced *SlMBP21* mRNA generated enlarged and fused sepals, while simultaneous inhibition of *SlMBP21* and *SlMADS1* led to larger (longer and wider) and fused sepals than that in *SlMBP21*-RNAi lines. qRT-PCR results exhibited that the transcripts of genes relating to sepal development, ethylene, auxin and cell expansion were dramatically changed in *SlMBP21*-RNAi sepals, especially in *SlMBP21*-*SlMADS1*-RNAi sepals. Yeast two-hybrid assay displayed that SlMBP21 can interact with SlMBP21, SlAP2a, TAGL1 and RIN, and SlMADS1 can interact with SlAP2a and RIN, respectively. In conclusion, *SlMBP21* and *SlMADS1* cooperatively regulate sepal development in tomato by impacting the expression or activities of other related regulators or via interactions with other regulatory proteins.

## 1. Introduction

MADS-box genes encode an important family of transcription factors that widely distributed in eukaryotes (i.e., plants, animals and fungi) [1]. In plants, this large gene family is divided into two super clades, the type I and the type II, on the basis of their evolutionary origin [2,3]. The MADS-box genes from type I lineage encode the SRF-like domain proteins that can be further classified into three subfamilies (Mα, Mβ, and Mγ) [4]. While the MADS-box genes of type II lineage which cover the most well-known MADS genes encode plant-specific MIKC-type proteins and MEF2-like proteins [5,6,7]. Type I MADS-box proteins only contain the MADS domain, whereas type II MADS-box proteins harbor four recognized domains (MADS (M), intervening (I), keratin-like (K) and C-terminal (C) domains) [5,8,9,10]. In the type II lineage, MIKC-type MADS-box proteins can be further classified into MIKC* and MIKC^C^ groups based on their different encoded I and K domains [11]. 

In plants, the most well-identified MADS-box transcription factors are the members of the MIKC^C^ group. They play essential roles in numerous critical biological processes of plant growth and development, including abiotic stress response, vegetative growth and reproductive development [12,13,14]. In some plant species, the MADS-box genes have been proved to be the important regulators for plants to adapt challenging environmental conditions. For example, *OsMADS26* controls various stress responses by regulating the expression of biosynthesis genes related to JA(jasmonic acid), SA (salicylic acid), ET (ethylene) and ROS (reactive oxygen species) in rice [15]. Overexpression of *SlMBP11* in tomato enhanced tolerance to salt stress by reducing the relative electrolyte leakage and MDA content, and increasing the water and chlorophyll content in transgenic plants [16]. SlMBP8 function as a negative stress-responsive transcription factor in the drought and high-salinity stress signaling pathways by regulating multiple stress-related genes [17]. 

In addition to the abiotic stress response, MADS-box transcription factors also function as the significant regulators in plant vegetative growth. Overexpression of *SlFYFL* in tomato delayed leaf and sepal senescence and fruit ripening, and affected abscission zone development [18]. Overexpressing the dominant negative form of *SlMBP20* resulted in leaf simplification and partly repressed the increase in tomato leaf complexity [19]. Up-regulation of *SlMBP11* in tomato led to decreased internode length, leaf size and plant height; increased node and leaf number; and excessively more branches were generated in each leaf axil [20]. In *Arabidopsis thaliana*, *agl21* mutants showed fewer and shorter lateral roots, while *AtAGL21* overexpression plants produced more and longer lateral roots; *AtAGL21* was found to positively regulate auxin accumulation in lateral root primordia and lateral roots by enhancing local auxin biosynthesis, thus stimulating lateral root initiation and growth [21]. Similarly, *OsMADS25* was reported to regulate root system development via auxin signaling in rice [22]. In tomato, four MADS-box genes, *MC* (*MACROCALYX*), *JOINTLESS*, *SlFYFL*, *SlMBP21*, are involved in regulating abscission zone development [18,23,24,25]. 

Nowadays, numerous MADS-box genes have been proved to play crucial roles in regulating inflorescence development, such as *PAP2* (*PANICLE PHYTOMER2*), *OsMADS14*, *OsMADS15*, and *OsMADS18* in rice [26], *MC* (*MACROCALYX*), *JOINTLESS*, *SlMBP21*, *SlCMB1*, *SlMADS1* in tomato [27,28,29,30], and four SEPALLATA (SEP) MADS-box genes (*SEP1–4*) in *Arabidopsis* [31,32], and so on. These MADS-box genes generally regulate inflorescence development by regulating the expression of genes related to floral meristem development or hormone synthesis. Moreover, MADS-box transcription factors act as the important regulators in plant parthenocarpy, cuticle development, flowering time, fruit development and ripening, embryo and seed development and floral organ identity determination, etc. For instance, down-regulation of the MADS-box gene (*TM29*) or mutation of *SlAGAMOUS-LIKE 6* (*SlAGL6*) in tomato lead to parthenocarpic fruit development [33,34]; tomato MADS-box gene *TAGL1* (*TOMATO AGAMOUS-LIKE1*) regulates cuticle development by participating in the transcriptional control of cuticle development mediating the biosynthesis of cuticle components [35,36]. Five MADS-box genes, *SOC1* (*SUPPRESSOR OF OVEREXPRESSION OF CONSTANS1*), *AGL24* (*AGAMOUS-LIKE GENE 24*), *FLC* (*FLOWERING LOCUS C*), *AP1* (*APETALA1*) and *SVP* (*SHORT VEGETATIVE PHASE*), are required for flowering time in *Arabidopsis* [37,38,39,40,41,42]. *AtAGL24* acts as a promoter of flowering in *Arabidopsis* and is positively regulated by vernalization [42]. The rice plants with increased *OsMADS18* levels or overexpressing the MADS-box gene *UNS* (*UNSHAVEN*) in petunia results in an advanced flowering time, respectively [43,44]. MADS-box genes also act as vital regulators in regulating fruit ripening, and these genes generally function at key control sites in the regulatory network of fruit ripening or ethylene synthesis. For example, RIN, a tomato MADS-box transcription factor, is a significant regulator of fruit ripening, in mutant inhibited fruit ripening with a failure to fruit ripening, including the reduced respiratory climacteric, associated ethylene evolution, and carotenoid accumulation, suppressed softening and production of flavor compounds [29]. In addition to RIN, other MADS-box genes are also involved in regulating tomato fruit ripening, such as *FUL1*(*FRUITFULL1*), *FUL2* (*SlMBP7*), *TAGL1*, *SlMADS1* and *SlFYFL* [18,45,46,47].

The regulation of floral organ identity is the most notable role for MADS-box genes in flowering plants. Numerous MADS-box genes are identified as five classes (A, B, C, D and E) following their functions in floral organ identity according to the ABCDE model. As described in this model, A-class genes are required for sepal and petal development, such as *APETALA1* (*AP1*) and *AP2* in *Arabidopsis* [48,49,50], and *MC* in tomato [29]; B-class genes regulate the development of petal and stamen, such as *PISTILLATA* (*PI*) and *AP3* in *Arabidopsis* [51,52], and *GLO* (*GLOBOSA*) and *DEF* (*DEFICIENS*) in *Antirrhinum majus* [53]; C-class genes refer to stamen and carpel, containing *AGAMOUS* (*AG*) in *Arabidopsis* [54]; class D genes specify ovule, including *FBP7* (*FLORAL B IND ING PROTEIN 7*) and *FBP11* in *Petunia* [55], and *STK* (*SEEDSTICK*), *SHATTERPROOF1* (*SHP1*) and *SHP2* in *Arabidopsis* [56]; and E class genes function in all four whorls of flowers through forming protein complexes with other classes, such as *SEP1–4* in *Arabidopsis* [31,32,57]. Moreover, many other MADS-box transcription factors that are involved in the development of floral organs also have been characterized. For instance, overexpression of *SlMBP11* in tomato exhibited shorter style, split ovary, polycarpous fruits and delayed perianth senescence [20], and the suppression of an SEP MADS-box gene, *SlCMB1*, generates enlarged and fused sepals due to the reduced expression of sepal-related genes (*MC*, *AP2a* and *GOBLET* (*GOB*)) [30]. Overexpressing *TM8* in tomato leads to the anomalous stamens with poorly viable pollen and altered expression of several floral identity genes, among them B-, C- and E-function ones [58]. After the ABCDE model of flower development, the floral quartet model was proposed by Theißen [59]. This model suggests that the identity of four different floral organs was determined by a tetramer of four floral homeotic proteins, respectively. The complex, which is composed of two class A proteins AP1 and two class E proteins SEP (2A + 2SEP), determines sepal identity; a complex harboring one AP1 protein, two class B proteins PI and AP3, and one SEP protein (A + 2B + SEP) specifies petal identity; a complex of one class C protein AG, two class B proteins PI and AP3, and one SEP protein (2B + C+SEP) determines stamen identity; a complex of two AG proteins and two SEP proteins (2C + 2SEP) specifies carpel identity; a complex of one AG protein, one SEP protein and one of each of D class proteins STK and/or SHP (C + 2D + SEP) determines ovule identity [59,60].

SEP MADS-box proteins play fundamental roles in the identity of floral organs and other biological processes. In *Arabidopsis*, the inner three whorls of floral organs (petal, stamen, and carpel) developed into sepals in *sep1sep2sep3* triple mutants, and the indetermination of flower development was observed [32]. While in *sep1sep2sep3sep4* quadruple mutants, the four whorls of floral organs were replaced by vegetative leaves, and flower development become more indeterminate [31]. In tomato, six SEP MADS-box transcription factors (TM5, TM29, SlCMB1, SlMBP21, RIN, and SlMADS1) have been identified to regulate different biological processes by protein interaction and/or gene expression regulation [24,29,30,33,47,61,62,63,64]. For instance, SlCMB1 is involved in fruit ripening, inflorescence and sepal development [30,63]; SlMBP21 plays essential roles in the development of an abscission zone, inflorescence, and sepals [24,27,64]; SlMADS1 regulates fruit ripening and inflorescence development [27,47]; and the down-regulation of *TM29* results in floral reversion and parthenocarpic fruit development [33]. The sepals is the outermost whorl of floral organs, and protects flower buds and young fruits. Normal sepal development is crucial for successful reproductive development. As the E class genes, the SEP MADS-box transcription factors are involved in the sepal identity in the ABCDE model and the quartet model during floral organ development [60,65]. In *Arabidopsis*, four SEP MADS-box genes, *SEP1–4*, cooperatively specify floral organ identity [31,32]. The floral quartet model demonstrates that two class E SEP proteins are involved in specifying sepal identity [60].

In a previous study of our lab, SEP MADS-box gene *SlMADS1* (Solyc03g114840) negatively regulates tomato fruit ripening and its transcripts are primary accumulated in sepals [47]. Liu et al. reported that the antisense of *SlMADS1* resulted in elongated sepals in transgenic tomato flowers [24], and Soyk et al. reported that the *ej2* mutant with the loss of function of *EJ2* (i.e., *SlMADS1*) generates 50% longer sepals than that of WT [27]. Li et al. reported that the other one SEP MADS-box gene, *SlMBP21* (Solyc12g038510), regulates sepal size mediated via auxin and ethylene in tomato and it is primary expressed in flowers and sepals [64]. However, the functional connections between SlMBP21 and SlMADS1 are still unknown. So we speculate that *SlMBP21* and *SlMADS1* may cooperatively regulate sepal development in tomato. In this study, expression pattern analysis showed that the transcripts of *SlMBP21* were increased with flower and sepal development, and it was primarily expressed in sepal and pistil. The expression patterns of *SlMADS1* was similar to *SlMBP21* with the flower development, but *SlMADS1* transcripts decreased during sepal development and were primarily accumulated in sepals. This further suggested that both genes may regulate sepal development in tomato. Therefore, it is appealing to study whether or not *SlMBP21* and *SlMADS1* have cooperative functions in tomato sepal development. Using the RNAi method, tomato plants with reduced *SlMBP21* mRNA (*SlMBP21*-RNAi) generated enlarged and fused sepals, while the simultaneous inhibition of *SlMBP21* and *SlMADS1* (*SlMBP21*-*SlMADS1*-RNAi) led to fused and larger (longer and wider) sepals than that in *SlMBP21*-RNAi lines. These results suggested that *SlMBP21* and *SlMADS1* cooperatively play essential roles in sepal development. Then, we performed functional analysis of these two genes through the molecular and phenotypic characterization of single RNAi (*SlMBP21*-RNAi) and double RNAi (*SlMBP21*-*SlMADS1*-RNAi) silencing lines to deeply analyze the molecular mechanisms of the function of *SlMBP21* and *SlMADS1* in sepal development.

## 2. Results

### 2.1. Molecular Characterization of SlMBP21 and SlMADS1

To study the roles of SEP MADS-box proteins played in tomato, we concentrated our work on the two SEP MADS-box genes, *SlMBP21* and *SlMADS1*. Sequence analysis displayed that *SlMBP21* harbors a 753 bp ORF encoding a protein with 250 acid residues and this protein has an estimated molecular mass of 28.44 kDa (pI 8.95). *SlMADS1* possesses a 741 bp ORF encoding a protein with 246 acid residues, and SlMADS1 has an estimated molecular mass of 28.39 kDa (pI 8.84). Multiple alignment and phylogenetic analysis showed that both SlMBP21 and SlMADS1 proteins are SEP MADS-box proteins with four typical MADS-box domains (i.e., the MADS-box domain, the I domain, and the K-box domain) and their C-terminuses are highly different from other known MADS-box proteins (Figure 1A). Phylogenetic analysis displayed that these two MADS-box proteins belong to the SEP lineage and on the same branch (Figure 1B). SlMBP21 shares the highest similarity (74.2%) with SlMADS1 than other MADS-box proteins, suggesting that they may play similar functions in tomato development [63]. Moreover, SlMBP21 shares 63.7% similarity with SlCMB1 (a regulator of sepal development [30]), and SlMADS1 shares 66.8% similarity with SlCMB1 [30,63]. 

Moreover, promoter analysis showed that three ethylene-responsive elements (ERE motif, ATTTCAAA), one gibberellin-responsive element (P-Box, CCTTTTG) and one auxin-responsive element (TGA-box, TGACGTAA) were observed in the *SlMBP21* promoter region (Appendix A), and three ethylene-responsive elements (ERE motif, ATTTCAAA) and three gibberellin-responsive elements (two P-Box, CCTTTTG and one TATC-box, TATCCCA) were found in the *SlMADS1* promoter region (Appendix A). All these results indicate that *SlMBP21* and *SlMADS1* may play significant roles in tomato growth and development via the response to plant hormones.

### 2.2. Expression Profiles of SlMBP21 and SlMADS1 in Wild-Type Tomato Tissues

To study the underlying roles of *SlMBP21* and *SlMADS1* in tomato sepal development, we performed qRT-PCR (quantitative real time PCR) to detect the relative transcript levels of *SlMBP21* and *SlMADS1* in flowers and sepals at different developmental stages and in floral organs using specific primers (Appendix A) of these two genes. The results displayed that *SlMBP21* was highly expressed in sepals than other tissues and flowers and the transcripts of *SlMBP21* increased with flower and sepal development (Figure 2A–C). In floral organs, *SlMBP21* was mainly expressed in the sepal, petal and pistil, and lower expression was observed in the stamen (Figure 2D). *SlMADS1* had higher expression levels in sepals and flowers and it had similar expression patterns to *SlMBP21* with flower development, but the transcript levels of *SlMADS1* declined with sepal development (Figure 2E–G). In floral organs, *SlMADS1* transcripts were preferentially accumulated in sepals, and lower accumulation was observed in other floral organs (petal, stamen and pistil) (Figure 2H). These results suggest that both *SlMBP21* and *SlMADS1* may play significant roles in sepal development.

### 2.3. Generation of SlMBP21-RNAi Lines and SlMBP21-SlMADS1-RNAi Lines

To further investigate the functions of *SlMBP21* and *SlMADS1* in tomato development, we constructed the single RNAi vector of *SlMBP21* (*SlMBP21*-RNAi) (Appendix A) and double RNAi vector of *SlMBP21* and *SlMADS1* (*SlMBP21*-*SlMADS1*-RNAi) (Appendix A). These two RNAi vectors were transferred into wild-type tomato plants to obtain transgenic lines, respectively. In total, we obtained twelve single RNAi lines, in which *SlMBP21* was significantly diminished from twelve independent transgenic lines, and three lines (SRNAi02, SRNAi05, and SRNAi08) which showed the lowest transcript levels of *SlMBP21* were used for further investigation (Figure 3A). For the double RNAi lines (*SlMBP21*-*SlMADS1*-RNAi), we obtained eleven independent transgenic lines, and three lines (DRNAi06, DRNAi07, DRNAi09) which simultaneously showed the lowest transcript levels of *SlMBP21* and *SlMADS1* were used for further investigation (Figure 3B).

### 2.4. Double SlMBP21-SlMADS1-Silenced Lines Generated Larger Sepals Than the Single SlMBP21 Silencing Lines

During the developmental process of tomato plants, the anthesis time was recorded. In addition to abolished development of the tomato flower abscission zone in *SlMBP21*-RNAi and *SlMBP21*-*SlMADS1*-RNAi lines (Figure 4A,B), we also noticed that the sepals of single *SlMBP21* silencing lines grown faster than the wild type. The *SlMBP21*-RNAi sepals showed larger size than the wild type at the same stage (Figure 3A). Meanwhile, the *SlMBP21*-*SlMADS1*-RNAi lines showed faster sepal growth than the single *SlMBP21* silencing lines and WT (Figure 3A), and as result, the sepal size of double *SlMBP21*-*SlMADS1* silencing lines was larger than the single *SlMBP21* silencing lines at the same stage (Figure 4B,C). Moreover, part of the double RNAi lines, which harbored the strongly simultaneous suppression (more than 90%) of *SlMBP21* and *SlMADS1,* generated larger and leafy sepals than other double RNAi lines which harbored the weak suppression (70–80%) of *SlMBP21,* and *SlMADS1* produced only enlarged but not leafy sepals (Figure 4C). The increased sepal size with the decreased expression levels of these two genes in the double RNAi lines was observed. These results indicated that the sepal size was impacted by different suppression of *SlMBP21* and *SlMADS1* expression levels.

In view of the enlarged sepal size in single and double RNAi lines compared with WT, some metrical data of wild-type and transgenic sepals at 3 DPA (day post-anthesis) and 15 DPA after anthesis were obtained. On the third day (3 DPA) after anthesis, the *SlMBP21*-RNAi sepals were about 10 mm longer than the wild-type, and the double *SlMBP21*-*SlMADS1* silencing sepals were about 12 mm longer than the *SlMBP21*-RNAi sepals, while the double RNAi sepals were about 22 mm longer than the wild-type ones (Figure 5A). The 3 DPA sepal width of WT, single and double RNAi lines was also measured. The results showed that the *SlMBP21*-RNAi sepals were about 1.0–1.6 mm wider than WT, and the *SlMBP21*-*SlMADS1* silencing sepals were about 0.8–2.1 mm wider than *SlMBP21*-RNAi sepals, while the double RNAi sepals were approximate 2.2–3 mm wilder than the wild-type (Figure 5B). Twelve days later (15 DPA), the *SlMBP21*-RNAi sepals still remained 10 mm longer than the wild-type, and the double *SlMBP21*-*SlMADS1* silencing sepals were approximate 11–18 mm longer than the *SlMBP21*-RNAi sepals, while the double RNAi sepals were about 20–26 mm longer than WT (Figure 5A). The width of *SlMBP21*-RNAi sepals was about 1–1.8 mm wider than the wild-type, and the double *SlMBP21*-*SlMADS1* silencing sepals were approximate 1.3–2.1 mm wider than the *SlMBP21*-RNAi sepals, while the *SlMBP21*-*SlMADS1* silencing sepals were about 2.7–3.1 mm wider than WT (Figure 5B). These results indicate that both *SlMBP21* and *SlMADS1* synergistically control tomato sepal development.

### 2.5. Both Single SlMBP21 and Double SlMBP21-SlMADS1 Silencing Sepals Fused Together and Generated a Leaf-like Structure

In addition to enlarged sepals, the overwhelming majority of flowers of the single *SlMBP21* silencing lines and double *SlMBP21*-*SlMADS1* silencing lines generated fused sepals, in varying degrees, while the WT sepals can split completely with each other (Figure 4A,B). Some sepals of single and double RNAi lines fused together, however, even after the flower opened completely. The transgenic petals failed to unfold normally on account of the constraint of fused sepals. These fused sepals of single *SlMBP21* silencing lines and double *SlMBP21*-*SlMADS1* silencing lines could be easily separated by hand. We performed the statistical analysis related to the sepals of WT, *SlMBP21*-RNAi and *SlMBP21*-*SlMADS1*-RNAi lines, and found that there was no difference in sepal number in one flower among them (Figure 5C), but the number of fused sepals in one flower and the ratio of fused flowers in one inflorescence were markedly higher in the transgenic lines than in WT (Figure 5D,E).

In consideration of the enlarged and fused sepals in single and double RNAi lines, the paraffin section of flowers (3 DPA) and sepals (15 DPA) was analyzed to study the differences among these three lines (WT, single RNAi lines and double RNAi lines). The results showed that the cross-section of the double silencing sepals were the widest, and the sepals of *SlMBP21*-RNAi lines were wider than WT (Figure 6A). In addition, we also found that the sepals of two kinds of RNAi lines generated a palisade tissue-like structure under the upper and lower epidermis, of which the phenotype was relatively more obvious in the *SlMBP21*-*SlMADS1* silencing sepals than the *SlMBP21*-RNAi lines (Figure 6A,B). Moreover, based on the cross section of 3 DPA flowers, almost all of the sepals of RNAi lines linked each other with few layers of cells while the wild-type sepals were separated (Figure 6C). These results suggested that *SlMBP21* and *SlMADS1* are involved in the development of sepal cells.

### 2.6. Transcript Analyses of Genes Involved in Sepal Development

Since *SlMBP21* and *SlMADS1* were primarily expressed in sepals and flowers, and suppression of *SlMBP21* and *SlMBP21*-*SlMADS1* resulted in enlarged and fused sepals, the expression of known genes that are involved in sepal development (*SlCMB1*, *MC*, *SlAP2a* and *TAGL1*) and boundary establishment (*GOBLET*) were detected in sepals of WT, *SlMBP21*-RNAi and *SlMBP21*-*SlMADS1*-RNAi lines. Our results displayed that *SlCMB1*, *SlAP2a*, *TAGL1* and *GOBLET* were significantly decreased in these two kinds of transgenic lines, but no expression difference of *MC* was observed in transgenic sepals (Figure 7A–E).

In 2017, Li et al. reported that *SlMBP21* controls sepal size mediated by auxin and ethylene in Micro-Tom, so we also detected some genes related to ethylene (*SlACO1*, *ACS6*, *ERF1*), auxin (*SlAR-1*, auxin-repressed protein gene; *SlARP*, auxin-regulated protein gene; *SlABP19a*, auxin-binding protein gene ABP19a-like and *SlARF19*, *IAA3*, and *IAA9*) and cell expansion (*SlPEI*, pectinesterase inhibitor), according to their RNA-seq data in our study [64]. Our results displayed that the transcripts of *SlACO1*, *SlACS6*, *SlERF1*, *IAA3*, *IAA9* and *SlARF19* were significantly increased in *SlMBP21*-RNAi and *SlMBP21*-*SlMADS1*-RNAi sepals, while the expression levels of *SlAR-1*, *SlARP*, *SlABP19a* and *SlPEI* were dramatically reduced in these two transgenic lines, compared with WT (Figure 7F–O). It is noteworthy that the transcript levels of some of these fourteen detected genes were markedly changed in *SlMBP21*-*SlMADS1*-RNAi lines compared to that in *SlMBP21*-RNAi lines. Together, suppression of *SlMBP21* and *SlMADS1* had effect on the transcript levels of some genes related to sepal development.

### 2.7. SlMBP21 and SlMADS1 Can Interact with Other Sepal Development-Related Proteins

To study the interaction of SlMBP21 and SlMADS1 with other regulators in sepal development, SlAP2a, TAGL1, and the ripening-related protein RIN were selected to carry out the yeast two-hybrid assay [29,30,66,67]. The pGBKT7-*SlMBP21* and pGBKT7-*SlMADS1* vectors were generated to be used as the bait, respectively (Appendix A). The pGADT7-*SlMBP21*, pGADT7-*SlAP2a*, pGADT7-*TAGL1* and pGADT7-*RIN* vectors were obtained to be used as the prey, respectively (Appendix A). In addition, self-activation of pGBKT7-*SlMBP21* and pGBKT7-*SlMADS1* was also performed and the results were negative (Appendix A). Figure 8 showed that the yeast grew on the QDO selective medium and turned blue on the QDO/X medium that contained the X-α-gal indicator, suggesting that SlMBP21 can interact with SlMBP21, SlAP2a, TAGL1, and RIN, while SlMADS1 can only interact with SlAP2a and RIN, and no interaction between SlMBP21 and SlMADS1 was observed. In addition, we performed the yeast two-hybrid assay again to verify the “positive control”, “negative control”, and the interactions of “pGBKT7-SlMBP21 & pGADT7-SlMBP21” and “pGBKT7-SlMBP21 & pGADT7-SlMADS1” (please see Appendix A).

## 3. Discussion

MADS-box proteins play crucial roles in various plant developmental processes: they are involved in the morphogenesis of almost all plant organs and throughout the whole plant life cycle, contributing to the overall life of plants [14]. Research on MADS-box genes holds potential significances for improving crop cultivation, increasing yield, and adapting plants to challenging environmental conditions. In previous investigations, it was extensively accepted that the SEP subclade is active in specifying floral organ identity, regulating fruit ripening and the development of pedicel abscission zone [31,32,33,62,64,68]. To date, six SEP MADS-box transcription factors, RIN, TM5, TM29, SlCMB1, SlMADS1 and SlMBP21, have been identified and characterized in tomato. RIN is the master regulator that controls fruit ripening of tomato plants, while TM5 regulates the differentiation of the petal, stamen and carpel [29,62]. Suppression of *TM29* results in floral reversion and parthenocarpic fruit development [33]. *SlCMB1* plays essential roles in fruit ripening, inflorescence architecture and sepal size [30,63]. *SlMADS1* is involved in inflorescence and sepal development and functions as a negative regulator during tomato fruit ripening [27,47]. *SlMBP21* controls the development of the pedicel abscission zone, inflorescence architecture and sepal size [24,27,64]. It also can be associated with enlarged sepals in SlMBP21 RNAi lines that can better protect inner floral organs and enhance photosynthesis and fruit set ratio [64]. So far, five classes (i.e., A, B, C, D and E) of MADS-box genes have been characterized to determine floral organ identities [69,70]. As E class genes, SEP MADS-box genes play crucial roles in all four whorls of floral organs following the ABCDE model [60,71]. For instance, previous studies have shown that four SEP MADS-box genes, *SEP1*-*SEP4*, redundantly regulate the development of floral organs in *Arabidopsis* [31,32]. Therefore, MADS-box genes, especially those in the SEP class, serve as essential regulators of floral organ identity, which provide crucial insights for plant genetics and genetic engineering.

Previous studies have reported that both *SlMBP21* and *SlMADS1* shared similar expression profiles as they were preferentially expressed in flowers and sepals [47,64]. As in our study, higher sequence similarity between SlMBP21 and SlMADS1 was observed on the amino acid level, and all these two SEP MADS-box proteins have the four typical MADS-box domains. The conservation of MADS-box domains across different plant species highlights their fundamental biological roles, aiding in phylogenetic and functional analyses. Moreover, the transcripts of *SlMBP21* and *SlMADS1* increased with flower development, but *SlMBP21* expression levels increased with sepal development while *SlMADS1* expression levels declined with sepal development. The transcripts of *SlMBP21* and *SlMADS1* were preferentially accumulated in sepals than the other three whorls of floral organs. The results here suggest that these two genes may be required for the identity of floral organs in tomato, especially for sepal development. 

Up to now, it has been reported that some MADS-box genes function redundantly and/or cooperatively during tomato development. For instance, two AGAMOUS clade MADS-box genes, *TAG1* and *TAGL1*, cooperatively control flower development and redundantly regulate sepal and pollen development in tomato [72]. *FUL1* (*FRUITFULL1*) and *FUL2* have a redundant function in regulating ethylene-independent aspects of fruit ripening in tomato [73]. Three MADS-box genes (*AtAG*, *AtSHP1*, *AtSTK*) redundantly control carpel and ovule development [56], and the other three closely related and functionally redundant MADS-box genes, *AtSEPALLATA1/2/3/4* (*AtSEP1/2/3/4*), are required for the development of petals, stamens and carpels in *Arabidopsis thaliana* [31,32]. Liu et al. reported that the antisense of *SlMADS1* in tomato results in elongated sepals [24] and Soyk et al. reported that the tomato *ej2* (i.e., *SlMADS1*) mutant generates 50% longer sepals than that of WT [27]. Suppression of *SlMBP21* resulted in longer sepals and suppressed pedicel abscission zone development in tomato [24,64]. Similarly, in this study, down-regulation of *SlMBP21* also resulted in suppressed pedicel abscission zone development and enlarged and fused sepals in Mill. cv. Ailsa Craig (AC^++^). The *SlMBP21*-RNAi sepals were significantly longer and wider than WT, while the *SlMBP21*-*SlMADS1*-RNAi plants displayed suppressed pedicel abscission zone development and a larger sepal size than that of WT and the single *SlMBP21*-RNAi lines. Interestingly, the *SlMBP21*-*SlMADS1*-RNAi lines produced different sepals due to the different suppression of *SlMBP21* and *SlMADS1* expression levels (weak and strong), and the sepal size increased with the decreased expression levels of *SlMBP21* and *SlMADS1*. Simultaneously, the cross-sectional area of 15 DPA sepals of *SlMBP21*-*SlMADS1*-RNAi lines was also wider than that of *SlMBP21*-RNAi lines and WT. Moreover, more of a palisade tissue-like structure in the upper and lower epidermis was observed in the 15 DPA sepals of the *SlMBP21*-*SlMADS1*-RNAi lines than that of the single RNAi lines, but not in the WT sepals. Combined with previous investigations and the results obtained in our study, we speculate that *SlMBP21* and *SlMADS1* may have cooperative functions in controlling sepal size.

Although many functions of MADS-box genes are well-understood in single gene study, their cooperative roles in specific biological processes, such as tissue development or stress responses, require more detailed studies. In the functional study of MADS-box genes, it can be found that silencing or knockout of one gene results in no or few weak phenotypic changes, while silencing or knockout of two or more genes generates the obvious phenotypic changes. And these phenotypic changes are significantly enhanced with the increased number of genes that are silenced or knocked out, such as the *TAG1*/*TAGL1*, *FUL1*/*FUL2*, *AtAG*/*AtSHP1*/*AtSTK*, and *AtSEP1*/*2*/*3*/*4* genes mentioned above [31,32,56,72,73]. In plants, some MADS-box genes can play different roles in various biological processes, while multiple genes can play the same role in the same biological process. However, what the relationship is between these genes that play a common function, and whether these genes have functional cooperation or redundancy are the important part of gene functional investigations. Therefore, simultaneously studying the functional relationship of two or multiple genes in the same biological process will be conducive to further analyzing the molecular mechanism of these genes in regulating this biological process, which will provide new research ideas and enlightenment for the investigation of a complex gene expression regulation network.

In tomato plants, many well-characterized genes have been reported to regulate sepal development, which contribute to the further investigation of sepal formation and development. For instance, the tomato A-class gene *MC* function as an important regulator in sepal development [29]. Down-regulation of an A-class gene, *SlAP2a*, led to longer and wider sepals [66]. Suppression of an E-class gene, *SlCMB1*, results in enlarged and fused sepals [30]. Tomato plants with increased *TAGL1* generate fleshy and succulent sepals which can ripen normally with fruit ripening [67]. Our study showed that *SlCMB1*, *SlAP2a* and *TAGL1* were markedly reduced in *SlMBP21*-RNAi and *SlMBP21*-*SlMADS1*-RNAi lines, and lower expression levels of these three genes were observed in *SlMBP21*-*SlMADS1*-RNAi lines. But another A-class gene, MC, had no significant expression changes both in WT and transgenic sepals. In *SlMBP21*-RNAi and *SlMBP21*-*SlMADS1*-RNAi lines, we also found that the number of fused sepals in each flower and the ratio of fused flowers in one inflorescence were markedly higher than in WT. The fused sepals linked each other with few layers of cells in cross section of flowers, indicating that the partial boundary morphogenesis of the first whorl was probably impacted. Previous research reported that tomato NAC transcription factor *GOBLET* is required for the inhibition of congenital fusion with primordia [74]. In our study, the transcripts of *GOBLET* were significantly reduced in *SlMBP21*-RNAi and *SlMBP21*-*SlMADS1*-RNAi sepals. Based on our results and previous studies, we come to a conclusion that *SlMBP21* and *SlMADS1* may cooperatively affect sepal development through directly and/or indirectly regulating the expression of A-function genes and control sepals fusion by impacting the transcripts of genes related to boundary establishment.

Previous RNA-seq data showed that the transcripts of genes related to auxin, ethylene and cell expansion were significantly impacted in *SlMBP21*-RNAi lines of Micro-Tom [64]. In our study, the expression levels of some cell expansion-, auxin- and ethylene-related genes (*SlACO1*, *SlACS6*, *SlERF1*, *IAA3*, *IAA9*, *SlARF19*, *SlAR-1*, *SlARP*, *SlABP19a*, and *PEI*) that were selected from their RNA-seq data were also dramatically influenced in the *SlMBP21*-RNAi and *SlMBP21*-*SlMADS1*-RNAi lines, and the transcript levels of some of them in the *SlMBP21*-*SlMADS1*-RNAi lines were higher/lower than that in *SlMBP21*-RNAi lines. Furthermore, the *SlMBP21* promoter region contains three ethylene-responsive elements (ERE motif, ATTTCAAA), one gibberellin-responsive element (P-Box, CCTTTTG) and one auxin-responsive element (TGA-box, TGACGTAA), and the *SlMADS1* promoter region contains three ethylene-responsive elements (ERE motif, ATTTCAAA) and three gibberellin-responsive elements (two P-Box, CCTTTTG and one TATC-box, TATCCCA) indicating the essential roles of *SlMBP21* and *SlMADS1* in responding to plant hormones. All these results demonstrate that *SlMBP21* and *SlMADS1* may act cooperatively in sepal formation and development through participating in plant hormone responsive and finally impact the development of sepal cells.

MADS-box transcription factors can form dimers or a multimer with other regulators to control plant growth and development [75,76,77,78]. Although the intricate interactions of MADS-box genes with numerous transcription factors pose challenges for understanding their functions, investigation of these interactions will help to further clarify the function of MADS genes in plant growth and development. For example, the AP1 orthologue MC controls the development of inflorescences, the abscission zone and sepals [23,29]. A recent study showed that SlMBP21 is implicated in regulating sepal size in tomato [64]. Suppression of *SlCMB1* results in altered inflorescence architecture and enlarged sepals [30]. Tomato plants with increased *TAGL1* transcript levels generate a fleshy sepal, which can ripen normally following fruits [67]. The MADS-box transcription factor, RIN, which is the master regulator of fruit ripening in tomato, regulates ethylene synthesis by increasing the transcript accumulation of ethylene signaling and ethylene biosynthesis genes (e.g., *ACS2* and *ACS4*) [79,80,81]. In addition to the MADS-box proteins mentioned above, *AP2a*, a transcription factor of the AP2 family, has been proven to regulate tomato sepal development [66]. In this study, the results of yeast two-hybrid assay showed that SlMBP21 can interact with SlMBP21, SlAP2a, TAGL1 and RIN, and SlMADS1 can interact with SlAP2a and RIN but not with TAGL1, but there was no interaction between SlMBP21 and SlMADS1. Moreover, our previous studies have demonstrated that the MADS-box transcription factor SlCMB1 which plays essential roles in the sepal development, can interact with MC, SlAP2a, SlMBP21, TAGL1 and RIN, respectively [30,63]. In the floral quartet model, the sepal identity is determined by a complex of two AP1 proteins and two AG MADS-box proteins (2A + 2SEP) [59,60], while in the ABCDE model, the sepal development is regulated by A and E class genes [60,71]. Leseberg et al. reported that SlMBP21 can interact with MC, but SlMADS1 can not interact with SlMBP21, MC, TAGL1 [75]. Furthermore, they also reported that both SlMBP21 and SlMADS1 can also interact with MADS-box proteins TM3, JOINTLESS, SlMBP24, and RIN, respectively [75]. So it is reasonable to believe that SlMBP21 and SlMADS1 may form tetramers with other functional proteins during tomato development. So we further speculate that SlMBP21 and SlMADS1 may form trimers or tetramers with other functional proteins (e.g., SlAP2a, RIN, SlCMB1, TAGL1, and MC) via protein bridges to control sepal development through the transcriptional regulation of other related genes.

## 4. Materials and Methods

### 4.1. Plant Materials and Growth Conditions

Tomato plants, *Solanum lycopersicum* (Mill. cv. Ailsa Craig, AC^++^), were used as the wild type (WT). The WT and transgenic tomato plants were grown in the standard greenhouse (25 °C, 16 h day and 18 °C, 8 h night cycle, 250 μmolm^−2^ s^−1^ light intensity and 80% humidity). The tomato flowers were tagged at anthesis. The DPA (day post-anthesis) was used to record the days of flower and sepal development. All samples were frozen with liquid nitrogen immediately and stored at −80 °C.

### 4.2. Total RNA Extraction and Sequence Analysis of SlMBP21 and SlMADS1

Total RNA was isolated from each sample using the RNAiso Plus (Takara, Dalian, China) following the manufacturer’s protocol. In the reverse transcription, 1 μg total RNA samples were used to synthesize the first-strand cDNA using the M-MLV reverse transcriptase (Promega, Madison, WI, USA) with oligo(dT)_20_ primer.

To clone the full length of *SlMBP21* and *SlMADS1*, 1–2 μL cDNA was used to employ the PCR reactions with primers *SlMBP21-Full-F*/*-R* and *SlMADS1-Full-F*/*-R*, respectively (Appendix A). After being tailed with a DNA A-Tailing kit (Takara, Dalian, China), the amplified products of *SlMBP21* and *SlMADS1* were linked into the vector pMD18-T (Takara, Dalian, China) to obtain *SlMBP21*-pMD18-T and *SlMADS1*-pMD18-T, respectively. Positive clones of each vector were picked out through *Escherichia coli* DH5α transformation and were conformed by sequencing. 

To analyze putative cis-elements in the *SlMBP21* and *SlMADS1* promoter region, promoter analysis was carried out using the promoter sequences (2800 bp regions upstream initiation codon ATG of gene-predicted ORF) of each gene on the Plant CARE website (http://bioinformatics.psb.ugent.be/webtools/plantcare/html/, accessed on 3 July 2023).

### 4.3. Construction of SlMBP21-RNAi and SlMBP21-SlMADS1-RNAi Vectors and Plant Transformation

In order to down-regulate *SlMBP21* expression, a RNAi vector was generated using PHANNIBAL and pBIN19 vector. A 381 bp specific DNA fragment of *SlMBP21* was amplified using *SlMBP21*-i-F/R primers (Appendix A). After being purified, the amplified fragments were digested with the estriction enzyme *Kpn* I/*Xho* I and *Hin*d III/*Xba* I, respectively. Then, the *SlMBP21*-RNAi vector (Appendix A) was generated using the same method following our previous study described [30].

In order to repress the expression of *SlMBP21* and *SlMADS1* simultaneously in tomato, the double RNAi vector of *SlMBP21* and *SlMADS1* was constructed using pDH51, PHANNIBAL and pBIN19 vector. Firstly, the 381 bp *SlMBP21*-specific DNA fragments and the 515 bp *SlMADS1* specific DNA fragments were amplified with the primers *SlMBP21-*1(p)-F/*SlMBP21*-1(X)-R and *SlMADS1*-1(X)-F/*SlMADS1*-1(B)-R (Appendix A), respectively. Secondly, after being digested with *pst* I and *Xba* I, the purified *SlMBP21* fragments were linked into the pDH51 plasmid at the *pst* I and *Xba* I restriction site to generate the pDH51-*SlMBP21* vector. Thirdly, the digested *SlMADS1* fragments using estriction enzyme *Xba* I and *BamH* I were cloned into the pDH51-*SlMBP21* plasmid at the *Xba* I/*BamH* I restriction site to yield pDH51-*SlMBP21-SlMADS1*. Fourthly, the START DNA polymerase (Takara) was used to amplify the tandem fragment of *SlMBP21* and *SlMADS1* using the primers *SlMBP21*-2(K, H)-F/*SlMADS1*-2(X, B)-R (Appendix A). Fifthly, the purified tandem fragments of *SlMBP21*-*SlMADS1* were digested by *Kpn* I/*Xho* I and *Hin*d III/*Bam*H I and linked into the pHANNIBAL vector at the *Kpn* I/*Xho* I and the *Hin*d III/*Bam*H I restriction site, respectively. At last, the double-stranded RNA expression unit, possessing the cauliflower mosaic virus 35S promoter, the sense-orientated *SlMBP21-SlMADS1* tandem fragment, PDK intron, the antisense-orientated *SlMBP21-SlMADS1* tandem fragment, and the OCS terminator, was digested with *Spe* I/*Sac* I and cloned into pBIN19 vector to generate the Pbin19-*SlMBP21-SlMADS1* vector (Appendix A).

The obtained binary plasmids of *SlMBP21* and *SlMBP21*-*SlMADS1* were transferred into *Agrobacterium tumefaciens* (strain LBA4404), respectively. After that, the transformation of tomato cotyledon explants mediated by *Agrobacterium tumefaciens* was performed to obtain *SlMBP21*-RNAi lines and *SlMBP21*-*SlMADS1*-RNAi lines [82]. The tissue culture plants were screened by kanamycin and the PCR reactions were carried out to select the positive transgenic plants with the *NPTII*-F/R primers (Appendix A).

### 4.4. qRT-PCR (Quantitative Real-Time PCR) Analysis

Total RNA extraction of each sample and reverse transcription were carried out following the method as described above. Then, the synthesized cDNAs were diluted to 15 ng/μL using RNase/DNase-free water. The qRT-PCR reactions of gene expression analysis were performed using SYBR Premix Go Taq (Promega, Madison, America) in a 10 μL total reaction volume (5 μL SYBR Premix Go Taq, 3 μL ddH_2_O, 0.25 μL each primer (10 mM), and 1.5 μL cDNA). No template control (NTC) and no reverse transcription control (NRT) experiments of each gene were performed. Tomato *SlCAC* gene was used as the internal standard [83]. The 2^−∆∆C^T method was used to analyze relative transcript levels of each gene [84]. All specific primers of genes used in this study were listed in Appendix A. Analyses of gene transcript levels were carried out in three independent biological repeats.

### 4.5. Anatomical Analyses of Flowers and Sepals

Flowers at 3 DPA and sepals at 20 DPA were fixed in FAA solution (50% absolute ethanol, 5% glacial acetic acid, 5% glycerin and 5% formalin) for 24 h. The paraffin section were performed following the same method of our previous study [30]. A microscope (OLYMPUS IX71, Tokyo, Japan) was used to observe the paraffin section of each sample.

### 4.6. Statistics of Fused Sepals and Sepal Length and Width

The number of fused sepals in a single flower and flowers with fused sepals of WT and transgenic lines were recorded. Sepal length and width of WT and transgenic (single and double RNAi) flowers were measured at 3 DPA and 20 DPA. In our statistics, the employed flowers were all the first and second flowers of each inflorescences. The 3 DPA flowers were used to observe the fused sepals. At least twenty flowers of each lines were contained in our statistics.

### 4.7. Yeast Two-Hybrid Assay

The yeast two-hybrid assay was carried out using the MATCHMAKER GAL4 Two-Hybrid System III following the manufacturer’s protocol (Clontech, Los Angeles, America). The ORFs of *SlMBP21*, *SlMADS1*, *SlAP2a*, *TAGL1* and *RIN* were amplified with primers *SlMBP21*(Y2H)F/R, *SlMADS1*(Y2H)F/R, *SlAP2a*(Y2H)F/R and *TAGL1*(Y2H)F/R, *RIN*(Y2H)F/R, respectively (Appendix A). The amplified products of *SlMBP21* and *SlMADS1* were linked into the *Eco*R I/*Bam*H I restriction site of the pGBKT7 vector to generate the pGBKT7-*SlMBP21* and pGBKT7-*SlMADS1* vector, respectively (Appendix A). The amplified products of *SlMBP21*, *SlAP2a*, *TAGL1* and *RIN* were inserted into the *Eco*R I/*Bam*H I restriction site of the pGADT7 vector to obtain the pGADT7-*SlMBP21*, pGADT7-*SlAP2a*, pGADT7-*TAGL1* and pGADT7-*RIN* (Appendix A), respectively. The pGADT7-*SlMBP21*, pGADT7-*SlAP2a*, pGADT7-*TAGL1* and pGADT7-*RIN* vectors were transferred into the Y187 yeast strain and the pGBKT7-*SlMBP21* and pGBKT7-*SlMADS1* vectors were transferred into the Y2Hgold yeast strain, respectively. After that, the yeast two-hybrid assay was performed following the same method of our previous study [30].

### 4.8. Statistical Analysis

All obtained data were presented as mean ± standard deviation (± SD) in this study. Student’s *t*-test (*p* < 0.05, *p* < 0.01) was performed to analyze the significant difference between WT and transgenic lines. The measurement values indicate the mean values of three biological replicates.

## 5. Conclusions

Taken together, previous studies and our data from *SlMBP21*-RNAi and *SlMBP21*-*SlMADS1*-RNAi sepals prove that *SlMBP21* and *SlMADS1* cooperatively play significant roles in tomato sepal development through influencing the activities and/or expression of other regulators or via interacting with other regulatory proteins. Although the precise regulatory mechanisms of *SlMBP21* and *SlMADS1* in tomato sepal development remain to be discovered, these two MADS-box genes should be important targets for investigating the mechanisms underlying sepal development and also will provide a new tool for improving the yield of tomato and other crops. However, higher levels of regulatory cascades of sepal development-associated regulators still need to be discovered, for instance, the targets regulated directly or indirectly by *SlMBP21* and *SlMADS1* and the interactions of these regulators, which will probably be conducive to further studying the regulatory network of sepal development in tomato. Despite intensive research on some MADS-box family members, the functions of many MADS-box genes remain unexplored, hindering a comprehensive understanding. In summary, MADS-box genes play crucial roles in plant development, but their complexity and diversity present both challenges and opportunities for researchers in the field of plant biology.

## Figures and Tables

**Figure 1 ijms-25-02489-f001:**
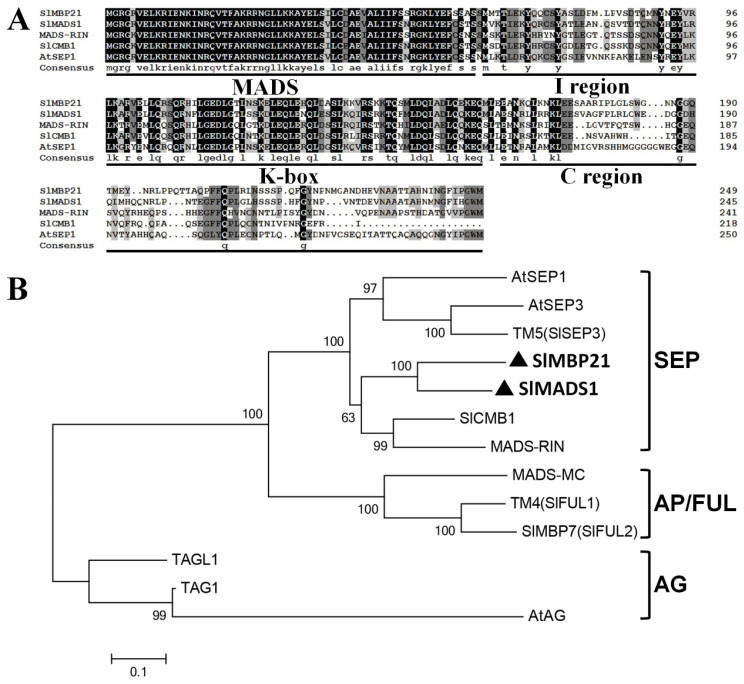
Sequence and phylogenic analysis of SlMBP21 and SlMADS1. (**A**) Multiple sequence alignment of SlMBP21 and SlMADS1 with other four known MADS-box proteins. The black part are the identical amino acids, and the gray are the similar amino acids. (The dark grey indicates the homology level between greater than or equal to 75 % and less than 100 %, and the light grey indicates the homology level between greater than or equal to 50 % and less than 75 %). The MADS-box, I region, K-box, and C region are identified. (**B**) Phylogenetic analysis of SlMBP21 and SlMADS1 with other known MADS-box proteins. SlMBP21 and SlMADS1 are highlighted with black triangles and bold fonts. Accession numbers of these proteins are listed as follows: AtSEP1 (AED92207.1), AtSEP2 (AEE73791.1), AtSEP3 (AEE30503.1), MADS-RIN (AF448522), SlMBP21 (NP_001275579), SlMADS1 (NP_001234380), TM5 (MADS5/LeSEP3) (NP_001234384/AY306153), MC (NP_001234665), LeFUL1 (AY098732.1), LeFUL2 (NM_001307938.1), TAG1 (L26295.1), TAGL1 (AY098735.2), AtAG (NP_001328877).

**Figure 2 ijms-25-02489-f002:**
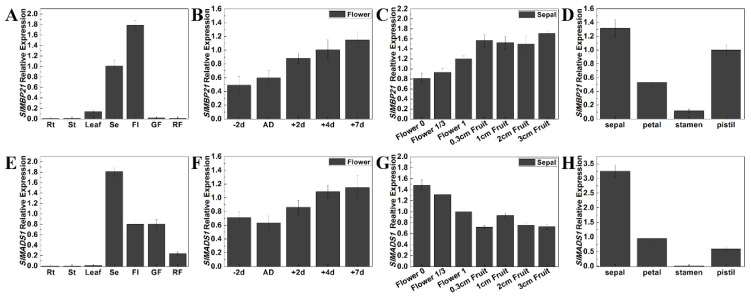
Expression analysis of *SlMBP21* and *SlMADS1* in different tissues of AC^++^ (Mill. cv. Ailsa Craig). (**A**) Relative expression of *SlMBP21* in different tissues of AC. Rt, roots; St, stems; Leaf; Se, sepal; FL, flowers at anthesis; GF, green fruits (25 DPA); RF, red fruits (7 days fruits after breaker). (**B**) Relative expression of *SlMBP21* in the different developmental stages of flowers. -2d, -2day-flowers before anthesis; AD, anthesis day; +2d, +2d-flowers after anthesis; +4d, +4d-flowers after anthesis; +7d, +7day-flowers after anthesis; DPA, days postanthesis. (**C**) Relative expression of *SlMBP21* in the different developmental stages of sepals. Flower 0, sepals of -2d-flowers before anthesis; Flower 1/3, sepals of flowers at anthesis; Flower 1, sepals of 2 days after anthesis; 0.3 cm Fruit, sepals of fruit (the diameter of fruit is 0.3 cm); 1 cm Fruit, sepals of fruit (the diameter of fruit is 1 cm); 2 cm Fruit, sepals of fruit (the diameter of fruit is 2 cm); 3 cm Fruit, sepals of fruit (the diameter of fruit is 3 cm). (**D**) Relative expression of *SlMBP21* in different floral organs of WT. (**E**) Relative expression of *SlMADS1* in different tissues of AC. (**F**) Relative expression of *SlMADS1* in the different developmental stages of flowers. (**G**) Relative expression of *SlMBP21* in the different developmental stages of sepals. (**H**) Relative expression of *SlMADS1* in different floral organs of WT. Error bars indicate SE.

**Figure 3 ijms-25-02489-f003:**
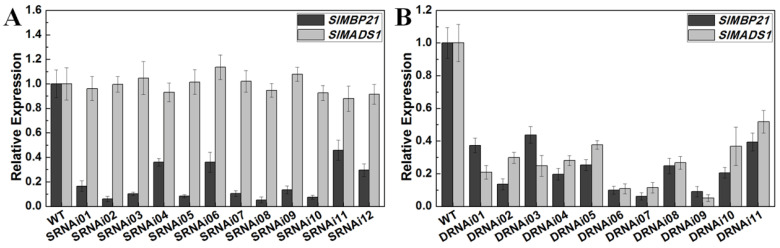
Relative expression of *SlMBP21* and *SlMADS1* in sepals of WT, *SlMBP21*-RNAi and *SlMBP21*-*SlMADS1*-RNAi lines. (**A**) Relative expression of *SlMBP21* and *SlMADS1* in sepals of WT and *SlMBP21*-RNAi lines at 3 DPA (day post-anthesis). (**B**) Relative expression of *SlMBP21* and *SlMADS1* in sepals of WT and *SlMBP21*-*SlMADS1*-RNAi lines at 3 DPA. Data are the means ± SE of three independent biological replicates.

**Figure 4 ijms-25-02489-f004:**
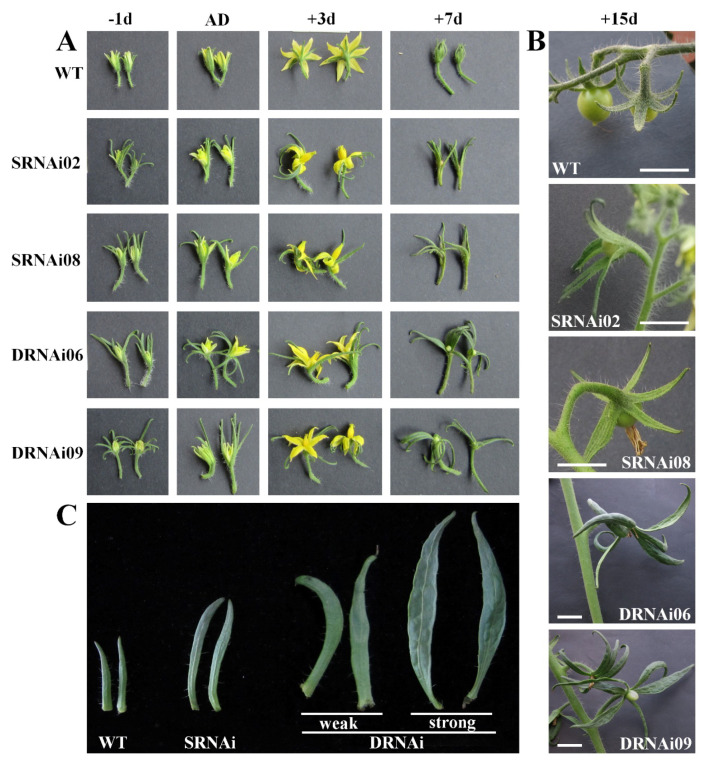
Sepal phenotypes in *SlMBP21*-RNAi and *SlMBP21*-*SlMADS1*-RNAi lines. (**A**) Sepals of WT, *SlMBP21*-RNAi and *SlMBP21*-*SlMADS1*-RNAi lines at different developmental stages; −1d, −1-day flowers before anthesis; AD, anthesis day; +3d, +3day-flowers after anthesis; +7d, +7-day flowers after anthesis; DPA, days post-anthesis. (**B**) Sepals of WT, *SlMBP21*-RNAi and *SlMBP21*-*SlMADS1*-RNAi lines at 15 DPA; SRNAi, *SlMBP21*-RNAi lines; DRNAi, *SlMBP21*-*SlMADS1*-RNAi lines; DPA, days post-anthesis. (**C**) Sepals of WT, *SlMBP21*-RNAi and *SlMBP21*-*SlMADS1*-RNAi lines at 20 DPA; weak, double RNAi lines which harbor weakly simultaneous suppression (70–80%) of *SlMBP21* and *SlMADS1*; strong, double RNAi lines which harbor strongly simultaneous suppression (more than 90%) of *SlMBP21* and *SlMADS1* generated larger and leafy sepals. Weak and strong representative images of phenotypic classes of sepals found in *SlMBP21*-*SlMADS1*-RNAi lines.

**Figure 5 ijms-25-02489-f005:**
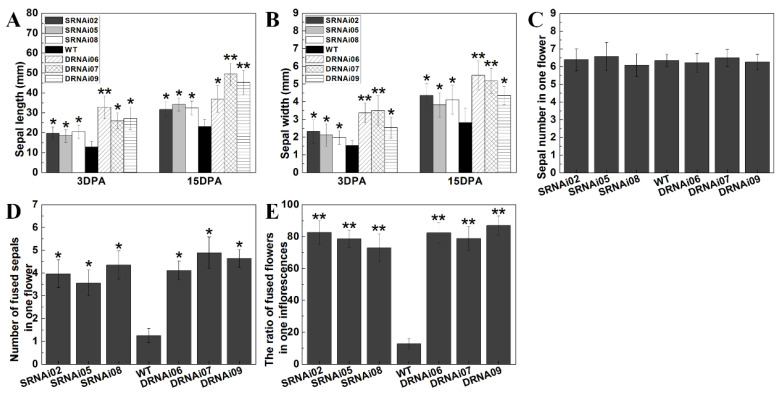
Statistical analysis of sepal in WT, *SlMBP21*-RNAi and *SlMBP21*-*SlMADS1*-RNAi lines. (**A**) The sepal length of WT, *SlMBP21*-RNAi (SRNAi) and *SlMBP21*-*SlMADS1*-RNAi (DRNAi) lines. (**B**) The sepal width of WT, *SlMBP21*-RNAi (SRNAi) and *SlMBP21*-*SlMADS1*-RNAi (DRNAi) lines. (**C**) The sepal number of one flower in WT, *SlMBP21*-RNAi (SRNAi) and *SlMBP21*-*SlMADS1*-RNAi (DRNAi) lines. (**D**) The number of fused sepals in one flower in WT, *SlMBP21*-RNAi (SRNAi) and *SlMBP21*-*SlMADS1*-RNAi (DRNAi) lines. (**E**) The ratio of fused flowers in one inflorescence in WT, *SlMBP21*-RNAi (SRNAi) and *SlMBP21*-*SlMADS1*-RNAi (DRNAi) lines. Data are the means ± SE of three independent biological replicates. The significant differences (*p* < 0.05, indicated by *) and highly significant difference (*p* < 0.01, indicated by **) were marked with the asterisks between WT and transgenic lines.

**Figure 6 ijms-25-02489-f006:**
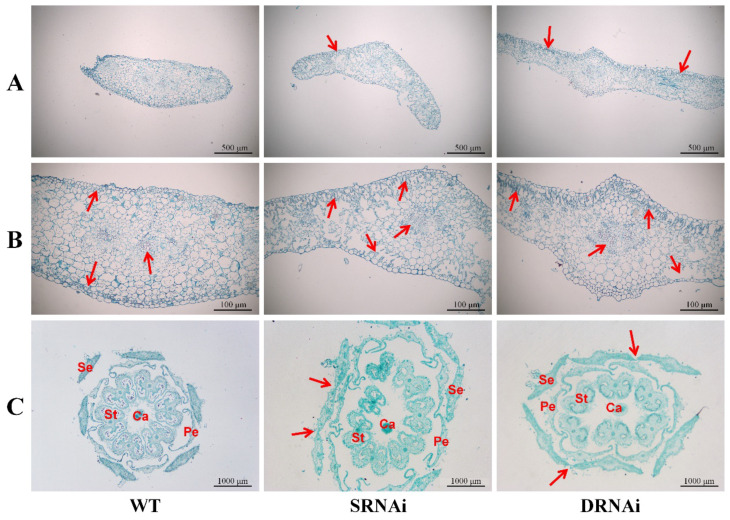
Morphological study of sepal and flower by paraffin section. (**A**) Comparison of sepal in WT, *SlMBP21*-RNAi (SRNAi) and *SlMBP21*-*SlMADS1*-RNAi (DRNAi) lines. Red arrows indicate the palisade tissue in sepals. The palisade tissue was generated in transgenic plants, while it was not found in sepals of WT. (**B**) Epidermal cell and vascular bundle cell of WT, *SlMBP21*-RNAi (SRNAi) and *SlMBP21*-*SlMADS1*-RNAi (DRNAi) lines. Red arrows indicate the palisade tissue and the vessel cells in sepals of transgenic plants. The palisade tissue was generated in transgenic plants, while it was not found in sepals of WT. The area and number of the vessel cells is larger than that in WT. (**C**) The flower (3DPA) paraffin section of WT and transgenic lines. The connectives between two sepals were marked by the red arrows. DPA, day post-anthesis; Ca, carpel; St, stamen; Pe, petal; Se, sepal.

**Figure 7 ijms-25-02489-f007:**
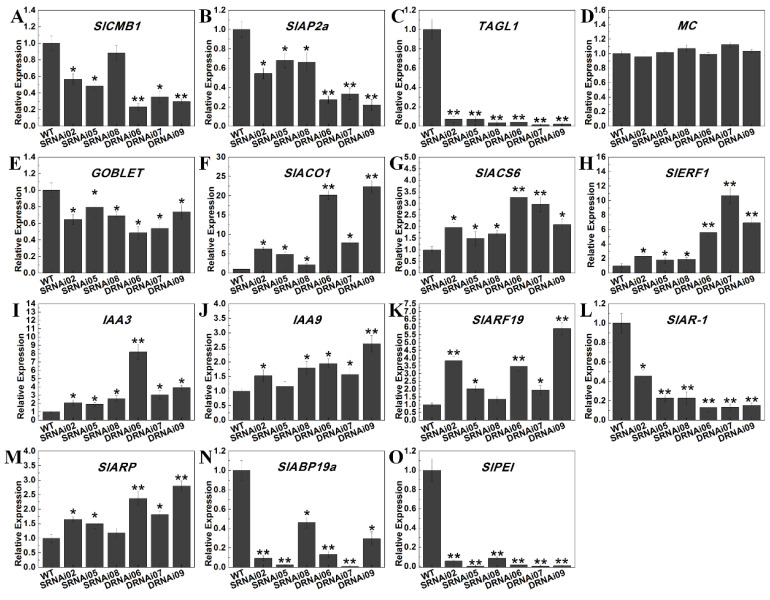
Relative expression of genes related to different biological processes during sepal development. (**A**–**D**). Relative expression levels of genes related to sepal development. (**E**) Relative expression level of gene related to boundary establishment. (**F**–**H**) Relative expression levels of genes related to ethylene. (**I**–**N**) Relative expression levels of genes related to auxin. (**O**) Relative expression level of gene related to cell expansion. The significant differences (*p* < 0.05, indicated by *) and highly significant differences (*p* < 0.01, indicated by **) were marked with the asterisks between WT and transgenic lines. Data are the means ± SE of three independent biological replicates.

**Figure 8 ijms-25-02489-f008:**
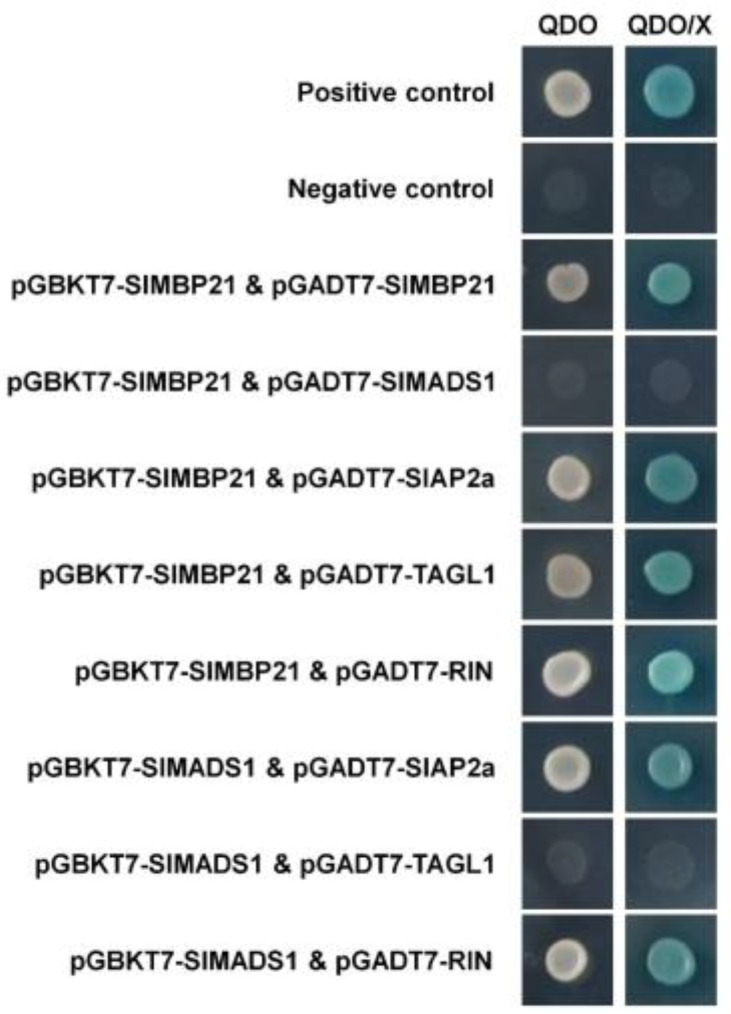
Yeast two-hybrid assay of SlMBP21 and SlMADS1 with other sepal development-related proteins. SlMBP21 could interact with SlMBP21, SlAP2a, TAGL1 and RIN but not with SlMADS1; SlMADS1 could interact with SlAP2a and RIN, but not with TAGL1, individually. QDO, SD medium lacking Trp, Leu, His, and adenine. QDO/X, SD medium lacking Trp, Leu, His, and adenine with X-α-Gal.

## Data Availability

The data and materials supporting the conclusions of this study are included within the article.

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
