# Peer review of "Two SEPALLATA MADS-Box Genes, SlMBP21 and SlMADS1, Have Cooperative Functions Required for Sepal Development in Tomato"

_ijms, 2024, doi:10.3390/ijms25052489_

Round 1

Reviewer 1 Report

Comments and Suggestions for Authors

Comments and Suggestions for Authors

The present study aimed to investigate whether or not MADS box genes SlMBP21 and SlMADS1 have cooperative functions in tomato sepal development based on functional analysis of these two genes through molecular and phenotypic characterization of single RNAi SlMBP21 RNAi) and double RNAi SlMBP21 SlMADS1 RNAi) silencing lines. MADS combination box transcription factors have crucial functions in multiple physiological and biochemical processes during plant growth and development. In the tomato MADS box gene family, SlMBP21 and SlMADS1 are important members of the SEPALLATA (SEP) clade. They perform essential functions, including regulation of inflorescence, sepal and abscission zone development, and fruit ripening. But their functions in tomato growth and development are still poorly known and need to be further investigated. As a result of the analysis, it was found that SlMBP21 and SlMADS1 genes function together in tomato sepal development.

The manuscript is composed according to Author guidlines of “International Journal of Molecular Sciences”. The methods used in the research are appropriate and sufficient to achieve the objectives of the study. The results are presented well and supported by tables and figures of good quality, and by statistical analysis.

The following recommendations can be made:

Abstaract

Add aim of the study and Material and methods.

The last sentence of the "Introduction": "Combined with previous studies and the results of our investigations, we conclude ..." is more appropriate as a conclusion.

Conclusion

The sentence: “Moreover, Li et al. … [63].” should be omitted from this section and moved to “Discussion”, in first paragraph, line 13: “SlMBP21 controls the development of pedicel abscission zone , inflorescence architecture and sepal size [23, 26, 63]. It also can be associated with the enlarged sepals in SlMBP21 RNAi lines that can better protect inner floral organs and enhance photosynthesis and fruit set ratio [63].”

In conclusion, this manuscript is recommended for publication in “International Journal of Molecular Sciences”.

Reviewer 2 Report

Comments and Suggestions for Authors

The article submitted for review deals with an important and timely topic. The importance of MADS-box genes in plant development is undisputed. In flowering plants, MADS-box genes act as homeotic selector genes determining floral organ identity and as floral meristem identity genes. By reviewing the current knowledge of MADS-box genes in ferns, gymnosperms, and several angiosperm species, we know that the phylogeny of MADS-box genes is strongly correlated with the origin and evolution of plant reproductive structures such as ovules and flowers. It is therefore likely that changes in MADS-box gene structure, expression, and function have been a major cause of innovations in reproductive development during land plant evolution, such as seed, flower, and fruit formation.

Overall, the paper is well-written and the results are significant. It should be published as it may be of interest to a wide range of readers in fields such as genetics, plant biology, and physiology. Before publication, it is important to complete and rewrite the introductory chapter of the article. At the beginning of the chapter, you should explain the function of MADS-box genes to readers who may not be specialists in plant genetics. I'd suggest that you try to summarise why these genes are so important for plant development and how they work. These genes may also be responsible for phenotypic plasticity in plants. To improve the quality of the article, please provide some general statements about the importance of this group of genes. I think this is necessary because the authors have given a lot of detailed information about the function of the MADS-box genes, which can be confusing without being an expert. Of course, the detailed information provided by the authors is valuable and should be retained. The results are interesting and not in doubt, and the discussion section is well and professionally written. Please check the text for linguistic errors, for example, in the abstract, replace 'tomatao' with 'tomato', ‘diplayed’ with ‘displayed’, etc. 

Comments on the Quality of English Language

The paper is written in good English. Only minor English language editing is required.

Reviewer 3 Report

Comments and Suggestions for Authors

Dear Authors,

The study, titled "Two SEPALLATA MADS-box genes, SlMBP21 and SlMADS1, have cooperative functions required for sepal development in tomato," focuses on the role of MADS-box genes, particularly the SEP class, in regulating sepal development in tomatoes. MADS-box genes are key regulators of various plant developmental processes, including flower, fruit, and root development, as well as responses to abiotic stress. In tomatoes, the genes SlMBP21 and SlMADS1, identified as members of the SEP class, were thoroughly examined for their role in sepal development. Gene expression analysis revealed that both SlMBP21 and SlMADS1 are predominantly expressed in sepals, with their expression levels changing throughout flower development. Functional studies using RNAi confirmed that downregulation of these genes leads to elongation of sepals. This suggests that SlMBP21 and SlMADS1 negatively regulate sepal development, aligning with previous research on SEP-class MADS-box genes in other plant species.

The paper highlights the versatility of MADS-box genes, their evolutionary conservation, and their significant relevance to plant genetics and genetic engineering for improving crop cultivation. It also underscores challenges related to analyzing complex interactions among MADS-box genes, the lack of detailed knowledge about certain family members, and the need for further research on their roles in various plant developmental processes.

Comments on the Paper:

1.     The abstract lacks clear structure, and conclusions in the abstract should be limited to the conducted research.

2.     The introduction should conclude with a clear research objective or the formulation of an alternative research hypothesis, leaving speculation for the hypothesis testing in the later stages of the study.

3.     Conclusions should not be stated at the end of the introduction; they belong in the "Conclusion" section.

4.     Results should be discussed according to the significance of the data.

5.     The discussion of results could be more in-depth.

6.     Conclusions should provide a summarizing and generalizing overview.

Strengths and Weaknesses:

A.    Strengths:

1.     Versatility of Roles: MADS-box genes play crucial roles in various plant developmental processes, contributing to the overall life of plants.

2.     Evolutionary Conservation: The conservation of MADS-box domains across different plant species highlights their fundamental biological role, aiding in phylogenetic and functional analyses.

3.     Regulation of Key Processes: MADS-box genes, especially those in the SEP class, serve as essential regulators of floral organ identity, providing crucial insights for plant genetics and genetic engineering.

4.     Relevance to Crop Production: Research on MADS-box genes holds potential significance for improving crop cultivation, increasing yield, and adapting plants to challenging environmental conditions.

B.    Weaknesses:

1.     Complex Interactions: The intricate interactions of MADS-box genes with numerous transcription factors pose challenges for understanding their functions.

2.     Lack of Detailed Knowledge about Many Family Members: Despite intensive research on some MADS-box family members, the functions of many genes in this family remain unexplored, hindering a comprehensive understanding.

3.     Potential Functional Diversity: The diversity of MADS-box genes suggests potential variations in functions between different plant species, complicating knowledge transfer.

4.     Less Understood Functions in Some Areas: While many functions of MADS-box genes are well-understood in certain contexts, their roles in specific processes, such as abiotic stress responses or vascular tissue development, require more detailed studies.

In summary, MADS-box genes play crucial roles in plant development, but their complexity and diversity present both challenges and opportunities for researchers in the field of plant biology.

Comments on the Quality of English Language

Minor English editing is required

Round 2

Reviewer 3 Report

Comments and Suggestions for Authors

Dear Authors,

I have reviewed the changes made to the manuscript, and I find that they are described extensively and seem to be well-directed responses to the reviewers' comments and suggestions. The revised text is coherent and understandable.
